# Long time thermal asymptotics of nonlinear Luttinger liquid from inverse scattering

Tom Price[1*]

**1** Institute for Theoretical Physics, Centre for Extreme Matter and Emergent Phenomena, Utrecht University, Princetonplein 5, 3584 CC Utrecht, the Netherlands
* t.a.price@uu.nl

October 2, 2018

## Abstract

I derive a Fredholm determinant for the thermal correlator of vertex operators with a conformally invariant density matrix and nonlinear time evolution. Using the method of nonlinear steepest descent I find the long time asymptotics at finite temperature. The asymptotics display exponentially small corrections in temperature to the finite temperature mobile impurity phenomenology.

# 1   Introduction

The behaviour of dynamical correlation functions in gapless one dimensional many body systems at zero temperature is now well established in terms of the mobile impurity model [1], with pioneering work [2, 3, 4] now over a decade old. Nevertheless, the problem of *thermal* dynamical correlation functions remains open [5, 6], and is of particular importance given experimental progress in measuring momentum resolved tunneling into one dimensional GaAs quantum wires [7, 8], and the dynamical structure factor of both Heisenberg antiferromagnets [9] and trapped cold atoms [10], where a finite temperature is an unavoidable fact of life. For integrable theories at zero temperature the mobile impurity phenomenology can give exact results [3, 11, 12, 13, 14]. On the other hand, at finite temperature, even for integrable systems the situation is rather less clear cut [5], in spite of remarkable progress from quantum inverse scattering [15, 16, 17, 6]. It is the purpose of this paper to give some exact results for the large $x, t$ asymptotics of a particular thermal Green function introduced in Refs. [18, 19]. I use the inverse scattering method [20] to evaluate the asymptotics of the classically integrable nonlinear Schrödinger equations driving the Green function, derived in Ref. [19]. In addition I give an explicit Fredholm determinant for the thermal Green function by observing that it obeys the correct equations of motion with correct initial data.

## 1.1   Definition of correlation functions in the nonlinear Luttinger liquid

In this paper I find asymptotics at large $x, t$ for thermal correlation functions of vertex operators. I begin by defining thermal correlation functions in a finite size system of length $L$,

$$
\bar{L}_\eta(x,t) = 2\pi \left(\frac{L}{2\pi}\right)^{-\eta^2} \mathrm{Tr}\, \rho e^{i\eta\phi^-(x,t)} \psi^\dagger(x,t) e^{i\eta\phi^+(x,t)} e^{-i\eta\phi^-(0,0)} \psi(0,0) e^{-i\eta\phi^+(0,0)},
$$

$$
G_\eta(x,t) = \left(\frac{L}{2\pi}\right)^{-\eta^2} \mathrm{Tr}\, \rho \;:\! e^{i\eta\phi(x,t)} \!::\! e^{-i\eta\phi(0,0)} \!:, \tag{1}
$$

$$
L_\eta(x,t) = 2\pi \left(\frac{L}{2\pi}\right)^{-\eta^2} \mathrm{Tr}\, \rho e^{i\eta\phi^-(x,t)} \psi(x,t) e^{i\eta\phi^+(x,t)} e^{-i\eta\phi^-(0,0)} \psi^\dagger(0,0) e^{-i\eta\phi^+(0,0)}.
$$

Vertex operators are defined in terms of normal ordered exponentials

$$
:\!e^{i\eta\phi(x)}\!:= e^{i\eta\phi^-(x)} e^{i\eta\phi^+(x)}, \tag{2}
$$

where the chiral boson has a mode expansion in terms of the currents $[J_n, J_{-m}] = m\delta_{nm}$,

$$
\phi^\pm(x) = -i \sum_{n \gtrless 0} \frac{1}{n} J_n e^{2\pi i n x/L}. \tag{3}
$$

The bosonization identity relates the fermion with anticommutator $\{\psi(x), \psi^\dagger(y)\} = \delta(x-y)$ and vertex operator

$$
\psi(x) = L^{-1/2} \;:\!e^{i\phi(x)}\!:\, e^{-2\pi i[\hat{N}+\frac{1}{2}]\frac{x}{L}} \hat{F}, \tag{4}
$$

where $\hat{N}$ measures the extra particles added and removed by $\hat{F}^\dagger$, $\hat{F}$ to the infinitely deep Fermi sea whose surface defines $k = 0$. Thus, up to $L$ dependent prefactors and zero modes, $L_\eta$, $\bar{L}_\eta$ are given by $G_{\eta\pm1}$. In the usual Luttinger liquid theory, the vertex operator with parameter $\eta$

arises in the right moving sector from diagonalization of the Luttinger Hamiltonian quadratic in bosons [21], and for the purposes of this paper we will consider it restricted to $\eta \in (-1/2, 1/2)$. It is also related to the phase shift at the Fermi point in models solvable by Bethe ansatz. We are able to focus only on excitations at a single (right) Fermi point since we shall use the Hamiltonian in the universal Imambekov–Glazman theory [18], Eqn. (5) below, which does not mix right and left movers.[1] At the right Fermi point, time evolution generated by a finite Fermi velocity $v$ is entirely accounted for by sending $x \to x - vt$, leaving the simplest possible nontrivial time evolution generated by the Hamiltonian

$$H = \frac{1}{3!m} \int_0^L :(\partial_x \phi)^3: \, \mathrm{d}x \, . \tag{5}$$

We do not include zero modes in the Hamiltonian, a choice to give the simplest form of nonlinear partial differential equations driving the correlation function. It is easy to include them to any results obtained at the end of the calculation. Although the Hamiltonian (5) is anharmonic in bosons, it describes free fermions (4) with quadratic dispersion. Meanwhile the trace in Eqns. (1) is over the Hilbert space of particle–hole excitations of the infinitely deep Fermi sea with a density matrix

$$\rho = \frac{1}{Z} \exp\left(-\beta \frac{v}{2} \int_0^L \mathrm{d}x \; :(\partial_x \phi)^2: \right). \tag{6}$$

As will become clear, the method is currently limited to the conformal density matrix (6). One can consider this density matrix an approximation to the canonical Boltzmann weight, or as a "generalized" Gibbs ensemble where we fix only the momentum of the right movers. When a nonlinear fermion dispersion is included in the density matrix the equal time correlation functions will not have a simple closed form and the approach we will take in this paper does not work. However, unlike the essential effect of nonlinearities on dynamics [1], the effect of dispersion in the thermodynamics should be nonsingular, at least for the correlation functions considered here.

The standard method of dealing with the double sum over Hilbert space, to account for the thermodynamic trace and dynamical time evolution in Eqns (1), is to pick a single microstate to represent the thermodynamic ensemble and knock out the trace over Hilbert space. The remaining sum over Hilbert space to describe time evolution can be dealt with using Fredholm determinants [20]. The same idea has been extended to nonequilibrium settings [24, 25]. Excitations over this representative state lead to the Yang–Yang equation and dressed energies. In contrast, here we keep the thermodynamic trace, a detail of note given the questions concerning the role of the dressed energies in the long time asymptotics [5].

The crucial feature of the density matrix quadratic in bosons (6) is the closed form of the Green functions (1) at $t = 0$, for example by analytic continuation of the finite size result

---

[1] Whether it is possible to extend the refermionization procedure used here, which can be traced to the work of [22, 23], to the case where correlation functions are taken in an entangled state of left and right movers, is an open problem.

$L \to -i\beta v$,

$$\bar{L}_\eta(x)/G_\eta(x) = \frac{2\pi i}{\beta v} e^{-\pi \frac{x}{\beta v}} \left(1 - e^{-2\pi \frac{x}{\beta v}}\right)^{2\eta-1},$$

$$G_\eta(x) = \left(e^{-i\frac{\pi}{2}} \frac{1 - e^{-2\pi \frac{x}{\beta v}}}{2\pi/\beta v}\right)^{-\eta^2}, \tag{7}$$

$$L_\eta(x)/G_\eta(x) = \frac{2\pi i}{\beta v} e^{-\pi \frac{x}{\beta v}} \left(1 - e^{-2\pi \frac{x}{\beta v}}\right)^{-2\eta-1}.$$

The power law as $T \to 0$ in $L_\eta$, $\bar{L}_\eta$ is a signature of the orthogonality catastrophe. As shown in Ref. [19], for density matrices $\rho$ quadratic in fermions, the functions $\Psi(x,t) \equiv L_\eta(x,t)/G_\eta(x,t)$, $\bar{\Psi}(x,t) \equiv \bar{L}_\eta(x,t)/G_\eta(x,t)$, obey the nonlinear Schrödinger equations (NLSEs)

$$i\partial_t \Psi + \frac{1}{2m}\partial_x^2 \Psi = \frac{1}{m}\eta^2 \Psi\bar{\Psi}\Psi,$$

$$-i\partial_t \bar{\Psi} + \frac{1}{2m}\partial_x^2 \bar{\Psi} = \frac{1}{m}\eta^2 \bar{\Psi}\Psi\bar{\Psi}. \tag{8}$$

These equations hold because of the generalized Wick's theorem [26]. By itself the system of nonlinear Schrödinger equations only determines the ratio of Green functions as $\eta$ is shifted by an integer. For translation invariant systems a third equation forms a closed set for $G_{\eta\pm1}(x,t)$ and $G_\eta(x,t)$ in the form of a Toda equation [19],

$$\partial_x^2 \log G_\eta(x,t) = -\eta^2 \bar{\Psi}\Psi. \tag{9}$$

At $t = 0$ the Toda equation can be checked using the explicit form of the correlation functions (7). In Ref. [27] a zero temperature scaling reduction of equations (8), (9) was derived using the Riemann–Hilbert methods of Ref. [20] and the scaling part of the Green function shown to be a tau function for the fourth Painlevé transcendent [28].

In Section 2 we use the differential equations (8) and $t = 0$ initial data to determine a Fredholm determinant representation for the Green functions (1) and then analyse its long time asymptotics in Section 3.

## 2 Fredholm determinant

The classical inverse scattering technique [29, 30] allows us to access the asymptotics of the nonlinear Schrödinger equations (8) given the correlators at $t = 0$, Eqn. (6). This leaves the problem of finding the (time dependent) integration constants when integrating up the Toda equation (9) to find $G_\eta(x,t)$. To this end it's useful to have a Fredholm determinant representation for $G_\eta(x,t)$. I will first state the result and then return to the proof. The Green function (1) with density matrix (6) and in the limit $L \to \infty$ is a Fredholm determinant

$$G_\eta(x,t) = \left(e^{-i\frac{\pi}{2}} \frac{\beta v}{2\pi}\right)^{-\eta^2} \sum_{n=0}^{\infty} \frac{1}{n!} \int_{-\infty}^{\infty} dk_1 \cdots \int_{-\infty}^{\infty} dk_n \; \det S(k_i, k_j)|_{i,j=1}^n$$

$$= \left(e^{-i\frac{\pi}{2}} \frac{\beta v}{2\pi}\right)^{-\eta^2} \det(1 + S). \tag{10}$$

The Fredholm kernel $S(k_i, k_j)$ takes the integrable form

$$S(k_i, k_j) = \frac{f_r(k_i)\sigma_3^{rs}g_s(k_j)}{k_i - k_j},$$ (11)

where $r, s = 1, 2$ and the components are

$$f_1(k) = g_2(k) = \frac{1}{\sqrt{e^{-\beta vk} + 1}} \frac{\Gamma\left(\frac{1}{2} + \eta - i\frac{\beta vk}{2\pi}\right)}{\Gamma(\eta)\Gamma\left(\frac{1}{2} - i\frac{\beta vk}{2\pi}\right)} e^{\frac{i}{2}\left[kx - \frac{k^2 t}{2m}\right]},$$

$$f_2(k) = g_1(k) = \frac{1}{\sqrt{e^{-\beta vk} + 1}} \frac{\Gamma\left(\frac{1}{2} + \eta - i\frac{\beta vk}{2\pi}\right)}{\Gamma(\eta)\Gamma\left(\frac{1}{2} - i\frac{\beta vk}{2\pi}\right)} e^{\frac{i}{2}\left[kx - \frac{k^2 t}{2m}\right]} Q(k + i0).$$ (12)

$$Q(z) = \int_{-\infty}^{\infty} \frac{dq}{z - q} \frac{1}{e^{\beta vq} + 1} \left[\frac{\Gamma\left(\frac{1}{2} - \eta + i\frac{\beta vq}{2\pi}\right)}{\Gamma(1 - \eta)\Gamma\left(\frac{1}{2} + i\frac{\beta vq}{2\pi}\right)}\right]^2 e^{-i\left[qx - \frac{q^2 t}{2m}\right]}.$$ (13)

A key object is the resolvent kernel $1 - R = (1 + S)^{-1}$, also integrable [20],

$$R(k_i, k_j) = \frac{F_r(k_i)\sigma_3^{rs}G_s(k_j)}{k_i - k_j},$$ (14)

with $F_r = (1 + S)^{-1} \cdot f_r$, $G_r = g_r \cdot (1 + S)^{-1}$. We apologise for the near collison in notation between the component of the resolvent kernel $G_s(k)$ and the Green function $G_\eta(x, t)$. In terms of the components of $R$ we can express ratios of Green functions

$$\eta^2 \frac{L_\eta(x, t)}{G_\eta(x, t)} = \int_{-\infty}^{\infty} dk \ F_1(k)g_2(k),$$

$$\frac{\bar{L}_\eta(x, t)}{G_\eta(x, t)} = -\int_{-\infty}^{\infty} dk \ F_2(k)g_1(k)$$

$$+ \int_{-\infty}^{\infty} dq \ \frac{1}{e^{\beta vq} + 1} \left[\frac{\Gamma\left(\frac{1}{2} - \eta - i\frac{\beta vq}{2\pi}\right)}{\Gamma(1 - \eta)\Gamma\left(\frac{1}{2} - i\frac{\beta vq}{2\pi}\right)}\right]^2 e^{-i\left[qx - \frac{q^2 t}{2m}\right]}.$$ (15)

This is the first result of this paper. The elements of the kernel contain square roots of the Fermi–Dirac occupations $\vartheta(k) = (e^{\beta vk} + 1)^{-1}$, a dynamical phase $e^{-i\theta} = e^{-i[k^2 t/2m - kx]}$, and "form factors"

$$\mathcal{F}(k) \equiv \frac{\Gamma\left(\frac{1}{2} + \eta - i\frac{\beta vk}{2\pi}\right)}{\Gamma(\eta)\Gamma\left(\frac{1}{2} - i\frac{\beta vk}{2\pi}\right)}, \quad \mathcal{G}(k) \equiv \frac{\Gamma\left(\frac{1}{2} - \eta + i\frac{\beta vk}{2\pi}\right)}{\Gamma(1 - \eta)\Gamma\left(\frac{1}{2} + i\frac{\beta vk}{2\pi}\right)}$$ (16)

that are the analytic continuation of the finite size matrix elements [31]

$$\langle e^{i\eta\phi^+(0)}\psi_p^\dagger\psi_{-q}\rangle = \frac{\mathcal{F}(p)\mathcal{G}(q)}{p + q}.$$ (17)

The proof is to observe [20] that the traces of the resolvent (15) of integrable kernels with time dependence of the form (12) satisfy NLSEs (8), so it remains only to check the value at

$t = 0$. This we can do by analytic continuation at $t = 0$ to the finite size result, discussed in detail in Appendix A, which explains the prescription $Q(k + i0)$ in the kernel. We note that an apparently similar determinantal representation of the correlation function (1) appears in Ref. [18], but differs being written as a grand canonical trace over many body states built over the empty vacuum, as is commonly used for full counting statistics [32]. In contrast the determinant here is in the Hilbert space of particle–hole excitations over the infinitely deep Fermi sea, and we are able to give an explicit form of the Fredholm kernel. A similar appearance of analytically continued $L \to -i\beta v$ form factors appears in Ref. [16] for the low energy limit of the delta Bose gas and Ref. [17] for the XXZ chain. It appears consistent with the results of Ref. [5] that the dynamical phase evolves with the energy relative to the ground state at zero temperature, not relative to a "representative state" with occupation $\vartheta(k)$. However, since we consider a free fermion model, there are no Bethe or Yang–Yang equations and we cannot say anything more regarding the role of the dressed dispersion in the asymptotics.

In spite of some effort I have not been able to find a *direct* proof of Eqn. (10) using the Lehmann representation and for now leave it as a challenge to any interested reader. To go from the Lehmann representation, which consists of two sums over Hilbert space with temperature $\beta$ entering only in the Boltzmann factor, to the single sum over Hilbert space with temperature entering the "form factors" $\mathcal{F}(k)$, $\mathcal{G}(k)$ of Eqn (16) appears a highly nontrivial summation identity. Apart from a direct attack it might be useful to try to find the finite size and finite temperature Fredholm kernel. Whether a closed form exists must be related to solvability of the forward scattering problem on the theta functions entering $L_\eta(x)/G_\eta(x)$, $\bar{L}_\eta(x)/G_\eta(x)$ at finite $L$ and $T$.

## 3 Large $x, t$ asymptotics

In this section I will derive the large $x, t$ asymptotics of the Green function (1) by finding an asymptotic solution to the Riemann–Hilbert problem, defined below, following the method of nonlinear steepest descent invented in Ref [30]. I set up the Riemann–Hilbert problem (RHP) and find the asymptotic solution at $T = 0$, before addressing the finite $T$ problem.

### 3.1 Riemann–Hilbert problem

In the standard way [20] define the following piecewise analytic matrix valued function of $z$

$$\Phi_{ij}(z) = \delta_{ij} - \int_{-\infty}^{\infty} \mathrm{d}k \; \frac{F_i(k)\sigma_3^{jk}g_k(k)}{z - k}. \tag{18}$$

The matrix $\Phi(z)$ tends to the identity at infinity and is analytic everywhere except across the real $z$ axis where it has a jump with boundary values $\Phi^\pm(k) = \Phi(k \pm i0)$ related by

$$\Phi^+(k) = \Phi^-(k)\left[\begin{pmatrix} 1 & 0 \\ 0 & 1 \end{pmatrix} + 2\pi i[1 - \vartheta(k)]\mathcal{F}(k)^2 e^{-i\theta}\begin{pmatrix} Q^+(k) & -1 \\ Q^+(k)^2 & -Q^+(k) \end{pmatrix}\right]. \tag{19}$$

If it exists, a solution to the RHP (19) with boundary conditions $\Phi(z) \to 1$ at infinity is unique. Expanding $\Phi(z)$ at infinity

$$\Phi(z) = 1 - \frac{1}{z}\int_{-\infty}^{\infty}\begin{pmatrix} F_1(k)g_1(k) & -F_1(k)g_2(k) \\ F_2(k)g_1(k) & -F_2(k)g_2(k) \end{pmatrix}\mathrm{d}k \; + \ldots \tag{20}$$

we see it contains all information on $L_\eta/G_\eta$, $\bar{L}_\eta/G_\eta$, and $\partial_x \log G_\eta$. With a view to finding an asymptotic solution to the RHP (19) it is convenient to make the following change of variables via the function $Q(z)$ of Eqn. (12),

$$m(z) = \Phi(z) \begin{pmatrix} 1 & 0 \\ Q(z) & 1 \end{pmatrix}, \tag{21}$$

which also tends to the identity at infinity and is analytic everywhere in $z$ except over the real axis. Here it has a jump that may be found using $Q_+(k) - Q_-(k) = -2\pi i \vartheta(k)\mathcal{G}(k)^2 e^{i\theta(k)}$, so the jump equation for $m(z)$ has the usual form familiar to classical inverse scattering

$$m^+(k) = m^-(k) \begin{pmatrix} 1 & \bar{R}(k)e^{-i\theta(k)} \\ R(k)e^{i\theta(k)} & 1 + \bar{R}(k)R(k) \end{pmatrix}, \tag{22}$$

with reflection coefficients

$$R(k) = -2\pi i \vartheta(k)\mathcal{G}(k)^2, \quad \bar{R}(k) = -2\pi i [1 - \vartheta(k)]\mathcal{F}(k)^2. \tag{23}$$

Given the transformation (21), the asymptotic behaviour of $\Phi(z)$ (20), and the connection of the elements of the Fredholm kernel and resolvent to the Green functions (10), (15), we can relate the Green functions to the residue of $m$ at infinity

$$m^{(1)} = \begin{pmatrix} -i\partial_x \log G_\eta(x,t) & \eta^2 L_\eta(x,t)/G_\eta(x,t) \\ \bar{L}_\eta(x,t)/G_\eta(x,t) & i\partial_x \log G_\eta(x,t) \end{pmatrix}. \tag{24}$$

## 3.2   Zero temperature asymptotics

Once $\mathcal{F}(k)$, $\mathcal{G}(k)$ are rescaled by factors of $(-i\beta v/2\pi)^{\pm 2\eta}$ so that we can take the limit as $T \to 0$, the Fermi–Dirac distributions entering the elements of the kernel (12) sharpen into step functions $\Theta(\pm k)$. The RHP becomes identical to that studied in [27] for the purpose of extracting a Painlevé equation, however, Riemann–Hilbert methods were not used to derive the asymptotics in that paper. We take this opportunity to show how this works. At large $t$ and with $k_* = mx/t$ fixed, the jump matrix is highly oscillatory and may be deformed, as in Fig. 1, such that it is exponentially close to the identity as $t \to \infty$ everywhere except in the vicinity of $k = 0$ (the Fermi point) and the saddle point $k = k_*$. At these points we can solve model Riemann–Hilbert problems that well approximate the solution at large $t$. In the vicinity of the Fermi point, $\mathcal{D}_<$, we define $m_<(z)$ as the solution to the RHP for the linear Luttinger liquid

$$m_<^+(k) = m_<^-(k) \begin{pmatrix} 1 & -2\pi i\Theta(k)\frac{1}{\Gamma(\eta)^2}k^{2\eta}e^{ikx} \\ -2\pi i\Theta(-k)\frac{1}{\Gamma(1-\eta)^2}(-k)^{-2\eta}e^{-ikx} & 1 \end{pmatrix}, \tag{25}$$

with $m_<(z) \to 1$ as $z \to \infty$. We take $\mathcal{D}_<$ a disc of radius $\Lambda_<$ such that the quadratic phase is negligible, i.e. $\Lambda_< \ll k_*$. This problem can be solved in terms of Whittaker functions [33, 34], but we shall not need the explicit form here. We only need the expansion at large $z$, Eqn. (24), which we have explicitly since the equal time Green function $G_\eta(x)$ is known. In the vicinity of the saddle point, $\mathcal{D}_>$, we define $m_>(z)$ that takes care of the jump over the real axis $\mathcal{C}_>$ that lies in $\mathcal{D}_>$. We take $\mathcal{D}_>$ a disc of radius $\Lambda_>$ such that the (suitably deformed) jump matrix is

exponentially small in $(z - k_*)^2 t/2m$ at the boundary $\partial\mathcal{D}_>$, which requires $\Lambda_> \gtrsim (t/m)^{-1/2}$. For $k_* > 0$, the jump is upper triangular

$$m_>^+(k) = m_>^-(k) \begin{pmatrix} 1 & -2\pi i \frac{1}{\Gamma(\eta)^2} k^{2\eta} e^{-i\theta(k)} \\ 0 & 1 \end{pmatrix} \tag{26}$$

and thus solved by Plemelj's formula

$$m_>(z) = \begin{pmatrix} 1 & \frac{1}{\Gamma(\eta)^2} \int_{\mathcal{C}_>} \frac{dk}{z-k} k^{2\eta} e^{-i\theta(k)} \\ 0 & 1 \end{pmatrix}, \tag{27}$$

while for $k_* < 0$ the jump is lower triangular

$$m_>^+(k) = m_>^-(k) \begin{pmatrix} 1 & 0 \\ -2\pi i \frac{1}{\Gamma(1-\eta)^2} (-k)^{-2\eta} e^{i\theta(k)} & 1 \end{pmatrix} \tag{28}$$

with solution

$$m_>(z) = \begin{pmatrix} 1 & 0 \\ \frac{1}{\Gamma(1-\eta)^2} \int_{\mathcal{C}_>} \frac{dk}{z-k} (-k)^{-2\eta} e^{i\theta(k)} & 1 \end{pmatrix}. \tag{29}$$

To leading order in large $t$, the solution to the RHP is given by $m_>(z) m_<(z)$ with an error tending to the identity as $t \to \infty$ [30]. From the solutions for $m_>$, $m_<$ we can read off the asymptotic behaviour of derivatives of the Green function using Eqn. (24). Since $m_>$ is traceless the only contribution to $\partial_x \log G_\eta$ is from the Fermi point. The same holds for $\partial_t \log G_\eta$, so that we have the seemingly unspectacular result

$$G_\eta(x,t) \sim G_\eta(x), \qquad x,t \to \infty, \eta \in (-1/2, 1/2). \tag{30}$$

This does not mean that there is no "fine structure" to the spectral function, merely that it does not govern the long time asymptotics of $G_\eta(x,t)$ for $\eta \in (-1/2, 1/2)$. For $k_* > 0$ we have leading contributions to $G_{\eta+1}(x,t)$

$$G_{\eta+1}(x,t) \sim \begin{cases} G_\eta(x,t) \frac{1}{\eta^2 \Gamma(\eta)^2} \int_{\mathcal{C}_>} dk\ k^{2\eta} e^{-i\theta(k)}, & x/t > 0, \\ G_{\eta+1}(x,0) & x/t < 0. \end{cases} \tag{31}$$

A saddle point evaluation of the integral yields explicit formulae for the large $x, t$ asymptotics for $\eta \in (-1/2, 1/2)$

$$G_{\eta+1}(x,t) \sim \begin{cases} e^{-i\frac{\pi}{2}[\frac{1}{2} - \eta^2]} \frac{\sqrt{2\pi}}{\Gamma(1+\eta)^2} k_*^{[\eta+1]^2} \left(\frac{k_*^2 t}{m}\right)^{-\eta^2 - \frac{1}{2}} e^{i\frac{k_*^2}{2m}t} & x/t > 0, \\ (-ik_* t/m)^{-[1+\eta]^2} & x/t < 0. \end{cases} \tag{32}$$

These asymptotics agree with the mobile impurity result [18] and the numerical solution to the Painlevé equation [27]. At $\eta = 0$ they reduce to the asymptotics of the free fermion. I will leave the study of larger $\eta$ for further work, where numerical solutions to the Painlevé IV transcendent governing the scaling solution at zero temperature [27] suggest the asymptotics continue to hold at larger $\eta$.

## 3.3 Finite temperature asymptotics

To begin with let us remind ourselves of the physics behind low temperature behaviour in the mobile impurity model [35]. When the temperature is much less than the energy of the impurity, the thermal smearing of the Fermi step does not affect the occupation of the impurity to exponentially accuracy and only the soft modes feel the temperature. In real space the low temperature criteria is that $T \ll vk_*$, which for given $T$ restricts us to sufficiently fast directions in $x, t$ plane. This is easy to see from the Fermi–Dirac factors in the jump matrix (22), where it is the criterion that we can separate the Fermi and saddle points. Riemann–Hilbert methods allow us to find large $x, t$ asymptotics at all $T$ as we now show. Fortunately the Riemann–Hilbert problem in the vicinity of the saddle point is familiar from standard classical inverse scattering method for the NLSEs, albeit without complex involution, and asymptotics have been studied [20, 29].

The first step is to define a new Riemann–Hilbert problem such that the jump matrix is exponentially small everywhere except near a few isolated points, where a model RHP can be solved that well approximates the exact solution at large $x, t$. We consider $x > 0$ so the saddle point is on the positive real axis and define the new RHP on the contour shown in Fig. 1 by defining the piecewise analytic matrix $n(z)$ in terms of the reflection coefficients $R(z)$, $\bar{R}(z)$ of Eqn. (23) that determine the jump matrix (22),

$$n(z) = m(z) \begin{pmatrix} 1 & -\bar{R}e^{-i\theta} \\ 0 & 1 \end{pmatrix}, \ z \in \text{I} \cup \text{V}, \quad n(z) = m(z) \begin{pmatrix} 1 & 0 \\ -\frac{R}{1+R\bar{R}}e^{i\theta} & 1 \end{pmatrix}, \ z \in \text{II},$$

$$n(z) = m(z) \begin{pmatrix} 1 & 0 \\ Re^{i\theta} & 1 \end{pmatrix}, \ z \in \text{IV} \cup \text{VI}, \quad n(z) = m(z) \begin{pmatrix} 1 & \frac{\bar{R}}{1+R\bar{R}}e^{-i\theta} \\ 0 & 1 \end{pmatrix}, \ z \in \text{III}.$$

(33)

In the remainder of the complex plane we take $n(z) = m(z)$. The slotted contour along the imaginary axis is necessary so that we avoid the poles in $R(z)$, $\bar{R}(z)$ in order that $n(z)$ is piecewise analytic. We use $e^{\pm i\theta}$ only in regions where it is exponentially small as $z \to \infty$, which means that $n(z) \to 1$ as $z \to \infty$. The result is that the jump problem for $n(z)$ tends to the identity as $t \to \infty$ everywhere except the saddle point $k_*$ and Fermi point $k_F$, and the real axis from $(k_*, \infty)$. The jump across the real axis is diagonal so can be removed by solving a scalar RHP.

Now we seek solutions to deal with the jumps in the vicinity of the Fermi and saddle points. In the vicinity of the Fermi point the jump contour wraps the imaginary axis. In the upper half plane the jump problem contains only $\bar{R}e^{-i\theta}$, and in the lower half plane only $Re^{i\theta}$. On the positive imaginary axis decay is governed by the real factor $e^{ikx}$ (recall we consider $x > 0$), and similarly on the negative imaginary axis decay is governed by $e^{-ikx}$. This means $n(z)$ will be close to 1 everywhere except a region of order $1/x$ near the Fermi point, where we may ignore the quadratic phase. The solution to this model RHP $n_<(z)$ is found by constructing $n_<(z)$ from $m_<(z)$ using the transformation (33) of the solution to the equal time RHP over the real axis

$$m_<^+(k) = m_<^-(k) \begin{pmatrix} 1 & -2\pi i[1 - \vartheta(k)]\mathcal{F}(k)^2 e^{ikx} \\ -2\pi i\vartheta(k)\mathcal{G}(k)^2 e^{-ikx} & 1 - (2\pi)^2\vartheta(k)[1 - \vartheta(k)]\mathcal{F}(k)^2\mathcal{G}(k)^2 \end{pmatrix}.$$

(34)

As for the zero temperature Fermi point, the solution is known from the mathematical literature in terms of hypergeometric functions [36], but just as at $T = 0$ we shall not need its explicit form. All is needed is its residue at infinity, which we already know from Luttinger liquid theory, Eqn. (7).

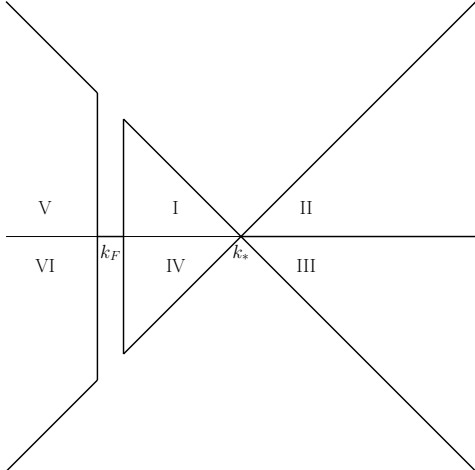

Figure 1: Deformation of the Riemann–Hilbert problem Eqn. (22) for $m(z)$ over the real axis (grey line) to a RHP on the solid contour for $n(z)$. The jump matrix for $n(z)$ is exponentially small at large $x, t$ everywhere except in a small region around the saddle point $k_*$, a small region around the Fermi point $k_F$, and a diagonal jump over $(k_*, \infty)$ on the real axis. The jump near the Fermi point on the slotted contour is necessary to avoid the poles in the reflection coefficients, Eqn. (23), used to define $n(z)$ in Eqn. (33).

In the vicinity of the saddle point, the RHP has the usual asymptotic solution in terms of parabolic cylinder functions [30]. The calculation is lengthy but known and yields a residue of $m_>$

$$
\begin{pmatrix}
\frac{1}{2\pi i}\int_{k_*}^{\infty} \mathrm{d}k \ \log\left[1 + \bar{R}(k)R(k)\right] & \bar{A}(k_*)\left(\frac{t}{m}\right)^{-\frac{1}{2}} e^{i\frac{k_*^2}{2m}t + i\nu \log\frac{k_*^2 t}{m}} \\
A(k_*)\left(\frac{t}{m}\right)^{-\frac{1}{2}} e^{-i\frac{k_*^2}{2m}t - i\nu \log\frac{k_*^2 t}{m}} & -\frac{1}{2\pi i}\int_{k_*}^{\infty} \mathrm{d}k \ \log\left[1 + \bar{R}(k)R(k)\right]
\end{pmatrix}. \tag{35}
$$

The reflection coefficients determine the exponent

$$
\nu = \frac{1}{2\pi}\log\left[1 + \bar{R}(k_*)R(k_*)\right], \tag{36}
$$

which is exponentially small in $\beta v k_*$ thanks to the product of particle and hole Fermi factors $\vartheta(k_*)[1 - \vartheta(k_*)]$. It also vanishes as $\eta \to 0$ like $\eta^2$ thanks to the Gamma functions in $\mathcal{F}(k_*)$, which guarantees we recover the free fermion results in this limit. The complex "amplitudes" are determined completely by the reflection coefficients

$$
\begin{aligned}
\bar{A}(k_*) &= \frac{1}{\sqrt{2\pi}} e^{i\frac{\pi}{4}} \Gamma(1 - i\nu) e^{2i\alpha(k_*)} e^{-\frac{\pi}{2}\nu} \bar{R}(k_*), \\
A(k_*) &= \frac{1}{\sqrt{2\pi}} e^{-i\frac{3\pi}{4}} \Gamma(1 + i\nu) e^{-2i\alpha(k_*)} e^{-\frac{\pi}{2}\nu} R(k_*),
\end{aligned} \tag{37}
$$

with the function $\alpha(k_*)$ defined

$$
\alpha(k_*) \equiv \frac{1}{2\pi}\int_{k_*}^{\infty} \log\frac{k - k_*}{k_*} \partial_k \log\left[1 + \bar{R}(k)R(k)\right] \mathrm{d}k . \tag{38}
$$

Replacing the Fermi weight by the zero temperature Fermi step $\vartheta(k) = 1 - \Theta(k)$, we indeed recover the residues from the triangular jump problems. Whilst the diagonal elements of $\mathrm{res}\,m_>$

are no longer zero, as in the zero temperature case, they will be subleading to the terms at the Fermi point, at least for sufficiently low temperatures. Thus we have for $\eta \in (-1/2, 1/2)$

$$G_\eta(x, t) \sim G_\eta(k_* t, 0). \tag{39}$$

$L_\eta(x, t)$ has oscillatory asymptotics on the supersonic side

$$L_\eta(x, t) \sim \begin{cases} G_\eta(k_* t, 0) \eta^{-2} \bar{A}(k_*) \left(\frac{t}{m}\right)^{-\frac{1}{2}} e^{i\frac{k_*^2}{2m}t - i\nu \log \frac{k_*^2 t}{m}}, & x/t > 0, \\ L_\eta(k_* t, 0), & x/t < 0. \end{cases} \tag{40}$$

Finally, $\bar{L}_\eta(x, t)$ has oscillatory asymptotics on the subsonic side,

$$\bar{L}_\eta(x, t) \sim \begin{cases} L_\eta(k_* t, 0), & x/t > 0, \\ G_\eta(k_* t, 0) A(k_*) \left(\frac{t}{m}\right)^{-\frac{1}{2}} e^{-i\frac{k_*^2}{2m}t + i\nu \log \frac{k_*^2 t}{m}}, & x/t < 0. \end{cases} \tag{41}$$

# 4 Conclusion

Using known differential equations driving the thermal correlation function of vertex operators with evolving quadratic fermionic dispersion [19], I used the nonlinear steepest descent method [30] to find long time asymptotics of the correlation function. At zero temperature this agrees with those known from the mobile impurity method. At finite temperature the asymptotics are still governed by the Fermi point and saddle point of the Riemann–Hilbert problem, which corresponds to the separation into Luttinger liquid and mobile impurity, but we find explicit corrections to the impurity correlation function from its free finite temperature form.

I also gave an explicit Fredholm determinant representation for the Green function, using the differential equations obeyed by the Green functions and conformal invariance at $t = 0$ to fix the initial conditions. This is possible due to the choice of a density matrix quadratic in bosons. Although when $t \neq 0$ the finite size and finite temperature correlators are not related by analytic continuation $L \to -i\beta v$, it would be interesting to find which finite $L$ properties (see e.g. Ref [37]) may be translated into finite temperature ones.

There are several directions to explore. The physical mechanism of the corrections to the impurity propagator is unclear. The question of whether there exists a closed form Fredholm determinant for both finite temperature and finite size, and nonlinear density matrix, also remains. The deformation of the RHP into an exponentially decaying one may also be useful for numerical evaluation of the Green functions, either attacking the RHP [38] or the Fredholm determinant directly [39]. As mentioned in the text, there is a summation identity lurking behind the Fredholm representation that may be new, or have some combinatorial interpretation analogous to the zero temperature connection to the $z$–measure. An extension to models with interacting spectra such as Lieb–Liniger or XXZ, rather than the free fermion spectra considered here, is required to understand the connection of the dressed energy in the thermodynamic Bethe ansatz and the oscillations in the large $t$ asymptotics [5]. See Ref. [6] for a review of recent work in this direction. It would also be interesting to try and extend the analysis to deal with multicomponent models to study spin–charge separation [34, 40]. Finally, I hope the strategy of combining a conformal density matrix with nonlinear time evolution can be helpful in finding explicit solutions in other problems.

# Acknowledgements

I am grateful for helpful discussions with Dmitry Kovrizhin and Dirk Schuricht.

**Funding information**   This work was supported by the Foundation for Fundamental Research on Matter (FOM), which is part of the Netherlands Organisation for Scientific Research (NWO), under Contract No. 14PR3168. This work is part of the D-ITP Consortium, a program of the Netherlands Organisation for Scientific Research (NWO) that is funded by the Dutch Ministry of Education, Culture and Science (OCW).

# A   Equal time sum rules

## A.1   Fredholm determinant: $G_\eta$

At $t = 0$ the determinantal expression for the Green function (10) with kernel Eqns. (11), (12) can be written

$$\det{(1+S)}|_{t=0} = \sum_{n=0}^{\infty} \frac{1}{n!} \int_{-\infty}^{\infty} \frac{e^{ik_1 x}\mathrm{d}k_1}{e^{-\beta v k_1}+1} \cdots \int_{-\infty}^{\infty} \frac{e^{ik_n x}\mathrm{d}k_n}{e^{-\beta v k_n}+1}$$
$$\times \det \left. \frac{\mathcal{F}(k_i)\mathcal{F}(k_j)Q^+(k_j) - \mathcal{F}(k_j)\mathcal{F}(k_i)Q^+(k_i)}{k_i - k_j} \right|_{i,j=1}^{n}. \tag{42}$$

In terms of $\mathcal{F}(k)$, $\mathcal{G}(k)$ of Eqn. (16), at $t = 0$

$$Q(z; x, t=0) = \int_{-\infty}^{\infty} \frac{\mathrm{d}q}{z-q} \mathcal{G}(q)^2 \frac{e^{-iqx}}{e^{\beta v q}+1}, \tag{43}$$

the kernel can be reexpressed

$$\frac{\mathcal{F}(k_i)\mathcal{F}(k_j)Q^+(k_j) - \mathcal{F}(k_j)\mathcal{F}(k_i)Q^+(k_i)}{k_i - k_j} = \int_{-\infty}^{\infty} \frac{e^{-iq_j x}\mathrm{d}q_j}{e^{\beta v q_j}+1} \frac{\mathcal{F}(k_i)\mathcal{G}(q_j)}{k_i - q_j + i0} \frac{\mathcal{F}(k_j)\mathcal{G}(q_j)}{k_j - q_j + i0}. \tag{44}$$

Pull out a factor of

$$\int_{-\infty}^{\infty} \frac{e^{-iq_j x}\mathrm{d}q_j}{e^{\beta v q_j}+1} \frac{\mathcal{F}(k_j)\mathcal{G}(q_j)}{k_j - q_j + i0} \tag{45}$$

from the $j$th column of the determinant, and since the determinant is antisymmetric in $k_i$, $q_j$ and summed over, we may write

$$\det{(1+S)}|_{t=0} = \sum_{n=0}^{\infty} \frac{1}{(n!)^2} \int_{-\infty}^{\infty} \prod_{l=1}^{n} \frac{e^{ik_l x}\mathrm{d}k_l}{e^{-\beta v k_l}+1} \int_{-\infty}^{\infty} \frac{e^{-iq_l x}\mathrm{d}q_l}{e^{\beta v q_l}+1}$$
$$\times \left( \det \left. \frac{\mathcal{F}(k_i)\mathcal{G}(q_j)}{k_i - q_j + i0} \right|_{i,j=1}^{n} \right)^2. \tag{46}$$

As always we consider $x$ having a positive imaginary infinitesimal. Then we close the contours over $k_i$ around the positive imaginary axis.[2] For $\eta \in (-1/2, 1/2)$, $\mathcal{F}(k)$ is pole free in the

---

[2]It is this step that cannot be performed when $t \neq 0$ as the exponent $e^{-ik^2 t/2}$ decays in opposite quadrants.

upper half plane, and with the $i0$ prescription we do not encounter any poles in the $k_i - q_j + i0$ denominator when deforming the $k_i$ integral to wrap the upper imaginary axis. Thus the only poles are the simple poles in the Fermi–Dirac distribution at $\beta v k_j = 2\pi i[m_j + 1/2]$. So

$$\det{(1+S)}|_{t=0} = \sum_{n=0}^{\infty} \frac{1}{(n!)^2} \sum_{m_1>0} \frac{2\pi i}{\beta v} e^{-2\pi[m_1+\frac{1}{2}]\frac{x}{\beta v}} \cdots \sum_{m_n>0} \frac{2\pi i}{\beta v} e^{-2\pi[m_n+\frac{1}{2}]\frac{x}{\beta v}}$$

$$\times \int_{-\infty}^{\infty} \frac{e^{-iq_1 x} \mathrm{d}q_1}{e^{\beta v q_1}+1} \cdots \int_{-\infty}^{\infty} \frac{e^{-iq_n x} \mathrm{d}q_n}{e^{\beta v q_n}+1} \left( \det \frac{\mathcal{F}\left(\frac{2\pi i}{\beta v}[m_i + \frac{1}{2}]\right) \mathcal{G}(q_j)}{2\pi i[m_i + \frac{1}{2}]/\beta v - q_j + i0} \right)^2. \tag{47}$$

Now we can deform the $q_j$ contours down around the lower imaginary axis, where again for $\eta \in (-1/2, 1/2)$ we collect poles at $q_j = -2\pi i[\bar{m}_j + 1/2]$. This lets us write $\det(1+S)|_{t=0}$ as

$$\sum_{n=0}^{\infty} \frac{1}{(n!)^2} \sum_{\substack{m_1,\cdots,m_n>0, \\ \bar{m}_1,\cdots,\bar{m}_n>0}} \left( \det e^{-\pi[m_i+\frac{1}{2}]\frac{x}{\beta v}} \frac{\mathcal{F}\left(\frac{2\pi i}{\beta v}[m_i + \frac{1}{2}]\right) \mathcal{G}\left(-\frac{2\pi i}{\beta v}[\bar{m}_j + \frac{1}{2}]\right)}{m_i + \bar{m}_j + 1} e^{-\pi[\bar{m}_j+\frac{1}{2}]\frac{x}{\beta v}} \right)^2 \tag{48}$$

which we can recognize as the Lehmann representation over particle–hole pair excitations of the zero temperature, finite size correlation function $\langle 0|e^{i\eta\phi^+(x)} e^{-i\eta\phi^-(0)}|0\rangle$, analytically continued $L \to -i\beta v$ [31]. On the other hand it's a standard calculation to find $\langle 0|e^{i\eta\phi^+(x)} e^{-i\eta\phi^-(0)}|0\rangle$, so we must have

$$\det{(1+S)}|_{t=0} = \left(1 - e^{-2\pi\frac{x}{\beta v}}\right)^{-\eta^2}. \tag{49}$$

The sum rule (49) appears as the normalization of the mixed $z$–measure [41]. In the first quantized approach to bosonization [42, 43] the vertex operator multiplies the ground state by the symmetric function $\prod_i \left(1 - e^{2\pi i[x_i - x]/L}\right)^{-\eta}$. The expansion of this function on free fermion eigenstates modulo the ground state, the Schur functions, appears in Ref. [44]. In the context of finite temperature, this summation identity appears in analysis of the critical form factors of Lieb–Liniger model [16] and XXZ chain [17].

## A.2  Fredholm minor: $L_\eta$, $\bar{L}_\eta$.

At zero temperature and for a finite system we can express $L_\eta(x,t)/G_\eta(x,t)$ in terms of the resolvent kernel $R$,

$$\frac{\langle \psi(x,t) e^{i\eta\phi^+(x,t)} e^{-i\eta\phi^-(0)} \psi^\dagger(0)\rangle}{\langle e^{i\eta\phi^+(x,t)} e^{-i\eta\phi^-(0)}\rangle} = \sum_{p,p'} \langle \psi(x,t) e^{i\eta\phi^+(x,t)} \psi_p^\dagger\rangle \left(\delta_{pp'} - R(p,p')\right) \langle \psi_{p'} e^{-i\eta\phi^-(0)} \psi^\dagger(0)\rangle. \tag{50}$$

By direct calculation we can express the matrix elements on the RHS in terms of the components $f_1$ of the finite size $S$ kernel,

$$\langle \psi(x) e^{i\eta\phi^+(x)} \psi_p\rangle = \frac{1}{\sqrt{L}} \eta^{-1} f_1(p). \tag{51}$$

This gives us the compact result

$$\eta^2 \frac{L_\eta(x,t)}{G_\eta(x,t)} = \frac{2\pi}{L} \sum_p F_1(p) g_2(p). \tag{52}$$

Likewise we can express $\bar{L}_\eta(x,t)/G_\eta(x,t)$ using

$$\frac{\langle\psi^\dagger(x,t)e^{i\eta\phi^+(x,t)}e^{-i\eta\phi^-(0)}\psi(0)\rangle}{\langle e^{i\eta\phi^+(x,t)}e^{-i\eta\phi^-(0)}\rangle} = \frac{1}{L}\sum_k \mathcal{G}(k)^2 - \frac{1}{L}\sum_k F_2(k)g_1(k). \tag{53}$$

These formulae appear in Ref. [45] for correlation functions taken in arbitrary coherent states. For completeness I sketch a direct proof for the ground state correlation functions. To prove Eqn. (50), begin by considering the momentum space correlation function

$$\frac{\langle\psi_k(t)e^{i\eta\phi^+(x,t)}e^{-i\eta\phi^-(0)}\psi^\dagger_{p'}\rangle}{\langle e^{i\eta\phi^+(x,t)}e^{-i\eta\phi^-(0)}\rangle} \tag{54}$$

and commute the fermions to the centre using the group–like property of the adjoint action of the vertex operator $e^{-i\eta\phi^\pm}$ [26]

$$e^{-i\eta\phi^+(x)}\psi_k e^{i\eta\phi^+(x)} = \sum_p \langle\psi_k e^{i\eta\phi^+(x)}\psi^\dagger_p\rangle\,\psi_p. \tag{55}$$

The matrix element restricts the summation to $p > 0$. Anticommuting the fermions in the transformed matrix element

$$\frac{\langle e^{i\eta\phi^+(x,t)}\psi_p\psi^\dagger_{p'}e^{-i\eta\phi^-(0)}\rangle}{\langle e^{i\eta\phi^+(x,t)}e^{-i\eta\phi^-(0)}\rangle} = \delta_{pp'} - \frac{\langle e^{i\eta\phi^+(x,t)}\psi^\dagger_{p'}\psi_p e^{-i\eta\phi^-(0)}\rangle}{\langle e^{i\eta\phi^+(x,t)}e^{-i\eta\phi^-(0)}\rangle} \tag{56}$$

we insert a resolution of the identity in between $\psi^\dagger_{p'}$, $\psi_p$ and recognize the first Fredholm minor of $S$,

$$S\begin{pmatrix}p\\p'\end{pmatrix} \equiv \sum_{n=0}^\infty \frac{1}{n!}\sum_{p_1\cdots p_n}\det\begin{pmatrix} S(p,p') & S(p,p_1) & \cdots & S(p,p_n)\\ S(p_1,p') & S(p_1,p_1) & \cdots & S(p_1,p_n)\\ \vdots & \vdots & \cdots & \vdots\\ S(p_n,p') & S(p_n,p_1) & \cdots & S(p_n,p_n) \end{pmatrix}. \tag{57}$$

A standard result of Fredholm theory says the ratio of the minor to the Fredholm determinant is equal to the resolvent $R$ of Eqn. (14). Taking a Fourier transform over $k,k'$ completes the proof of Eqn. (52). The relation (53) proceeds along similar lines. In real space

$$\frac{\langle\psi^\dagger(x,t)e^{i\eta\phi^+(x,t)}e^{-i\eta\phi^-(0)}\psi(0)\rangle}{\langle e^{i\eta\phi^+(x,t)}e^{-i\eta\phi^-(0)}\rangle}$$

$$= \frac{1}{\det(1+S)}\sum_{pp'}\langle\psi^\dagger(x,t)e^{i\eta\phi^+(x,t)}\psi_p\rangle\langle e^{i\eta\phi^+(x,t)}\psi^\dagger_p\psi_{p'}e^{-i\eta\phi^-(0)}\rangle\langle\psi_{p'}e^{-i\eta\phi^-(0)}\psi(0)\rangle \tag{58}$$

$$= \frac{1}{L}\sum_k\mathcal{G}(k)^2 - \frac{1}{\det(1+S)}\frac{1}{L}\sum_{kk'}\mathcal{G}(k')\langle e^{i\eta\phi^+(x,t)}\psi_{k'}\psi^\dagger_k e^{-i\eta\phi^-(0)}\rangle\mathcal{G}(k').$$

Concentrate on the last term. Insert a resolution of the identity between $\psi_{p'}$, $\psi^\dagger_p$, which becomes

$$\frac{1}{\det(1+S)}\frac{1}{L}\sum_{n=0}^\infty\frac{1}{n!}\sum_{p_0\cdots p_n}\det\begin{pmatrix} \sum_{k,k'}\mathcal{G}(k')\frac{\mathcal{F}(p_0)\mathcal{G}(k')}{p_0+k'}\frac{\mathcal{F}(p_0)\mathcal{G}(k)}{p_0+k}\mathcal{G}(k) & S(p_0,p_1) & \cdots & S(p_0,p_n)\\ \sum_{k,k'}\mathcal{G}(k')\frac{\mathcal{F}(p_1)\mathcal{G}(k')}{p_1+k'}\frac{\mathcal{F}(p_0)\mathcal{G}(k)}{p_0+k}\mathcal{G}(k) & S(p_1,p_1) & \cdots & S(p_1,p_n)\\ \vdots & \vdots & \cdots & \vdots\\ \sum_{k,k'}\mathcal{G}(k')\frac{\mathcal{F}(p_n)\mathcal{G}(k')}{p_n+k'}\frac{\mathcal{F}(p_0)\mathcal{G}(k)}{p_0+k}\mathcal{G}(k) & S(p_n,p_1) & \cdots & S(p_n,p_n) \end{pmatrix}. \tag{59}$$

The $j$th element of the first column is $g_1(p_j)f_2(p_0)$. The last step of the proof is to expand the determinant down the first column

$$\frac{1}{\det(1+S)} \frac{1}{L} \sum_{kk'} \mathcal{G}(k') \langle e^{i\eta\phi^+(x,t)} \psi_{k'} \psi_k^\dagger e^{-i\eta\phi^-(0)} \rangle \mathcal{G}(k) = \frac{1}{L} \sum_{p_0} g_1(p_0) f_2(p_0)$$

$$+ \frac{1}{\det(1+S)} \frac{1}{L} \sum_{n=0}^{\infty} \frac{1}{n!} \sum_{j=1}^{n} (-1)^j \sum_{p_0 \cdots p_n} f_2(p_0) \det \begin{pmatrix} S(p_0,p_1) & S(p_0,p_1) & \cdots & S(p_0,p_n) \\ \vdots & \vdots & \cdots & \vdots \\ S(p_{j-1},p_1) & S(p_{j-1},p_1) & \cdots & S(p_{j-1},p_n) \\ S(p_{j+1},p_1) & S(p_{j+1},p_1) & \cdots & S(p_{j+1},p_n) \\ \vdots & \vdots & \cdots & \vdots \\ S(p_n,p_1) & S(p_n,p_1) & \cdots & S(p_n,p_n) \end{pmatrix} g_1(p_j)$$

$$= \frac{1}{L} \sum_{p,p'} g_1(p) \left( \delta_{pp'} - R(p,p') \right) f_2(p'),$$

(60)

where the final line follows by reordering the columns of the determinant to recognize the expansion of the minor $S\begin{pmatrix} p_0 \\ p_j \end{pmatrix}$.

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
