# Peer review of "Long time thermal asymptotics of nonlinear Luttinger liquid from inverse scattering"

_SciPost Physics_

## Round 1 · Referee Report · Anonymous (Referee 1) · 2017-9-26

Strengths

The paper contains interesting results.

Weaknesses

The text is written very succinctly, therefore, it is difficult to follow the author's calculations.

Report

The manuscript deals with large space-time asymptotic behavior of thermal correlation functions of vertex operators. The author obtains Fredholm determinant representations for these correlations and applies the Riemann--Hilbert approach to evaluating their asymptotics.

The manuscript contains important and interesting results provided they are correct. However, the text is written so succinctly that it is difficult for the reader to follow the author's calculations. This remark especially concerns Section 3 on the RHP asymptotic analysis. In particular, I did not find an answer to the question of why the deformation of the jump contour is the same for the cases of zero and finite temperatures. This situation is very atypical. As a rule, these deformations are very different, which leads to different asymptotic behavior of the solution to the RHP. I would like to have an explanation of this atypical effect. Besides, I would like to understand why a vicinity of the Fermi point contributes to the asymptotics at finite temperature. This is quite natural in the case of zero temperature,
because the Fermi--Dirac distributions turn into the step-functions, leading to the non-analyticity in the jump-matrix in the Fermi point. However, for finite temperature, the jump-matrix is analytic in this point, thus, it is not obvious that it gives a contribution to the asymptotics of the solution to the RHP. Finally, turning back to the case of zero temperature, I should notice that strictly speaking, one has to describe the behavior of the RHP solution in the points of non-analyticity, otherwise the solution to the RHP is not unique. If this is not the case for the given RHP, it would be desirable to have some explanation.

I would like to have answers to the formulated questions before making a final decision on the manuscript. Concluding, I want to make few remarks concerning the notation and the citations.

I think that the author should give definitions of ALL the symbols used in equations, even if the meaning of some of them (like m, T, \beta) is intuitively clear for readers. Moreover, m's in Eq. (5) and Eq. (21) have different meaning.

To the best of my knowledge, Ref. [13] has nothing to do with the mobile impurity method, and it should rather be attributed to the QISM.

Ref. [16] deals with the low temperature limit, while there exists a paper arXiv:1011.3149 by the same authors where the case of a finite temperature is considered. Ref. [20] also should be attributed to the list of citations on the QUANTUM inverse scattering. Thus, the number of results obtained in this field becomes rather big, therefore, the first part of the sentence on line 9 of Introduction ( ‘...even for integrable systems the situation is rather less clear cut’) contradicts to the second part (‘ in spite of remarkable progress from quantum inverse scattering’).

Requested changes

  1. To answer the questions formulated in the report.
  2. To define all the symbols used in equations.
  3. To make corrections in the citation list.

  • validity: ok
  • significance: ok
  • originality: ok
  • clarity: low
  • formatting: good
  • grammar: excellent

Author:  Tom Price  on 2017-09-30  [id 177]

(in reply to Report 1 on 2017-09-26)

I thank the referee for their careful reading of the paper, comments on the literature, and thoughtful suggestions and questions. Here I answer their questions.

Deformation of the Riemann--Hilbert problem for arbitrary T occurs via the piecewise analytic transformation Eqn~(33), with regions of the complex plane labeled in Fig. 1. This includes zero temperature as a special, simpler case that is also treated separately in Section 3.2. The reflection coefficients $R, \bar{R}$ contain Fermi-Dirac factors and therefore have poles along the imaginary axis at nonzero T that merge into a cut when T=0. For $T < vk_*$ the slotted contour cannot be closed over the imaginary axis without encountering these poles, or the cut at T=0. This justifies the contributions from the Fermi point to the asymptotics even when T is nonzero, as is expected on physical grounds in finite temperature extensions of the mobile impurity model. Personally I find it encouraging that the asymptotics at low $T \ll uk_*$ approach those of zero temperature and would be interested to know of situations where this is not the case.

I chose to separate the zero and nonzero temperature calculations because at zero temperature the only RHP that needs to be solved, Eqns 26--27, is elementary. At nonzero temperatures the model RHP is not triangular and calculation is extremely lengthy, although a standard in the literature as it is just the asymptotics of the nonlinear Schrodinger equation when no nonanalyticities are encountered in the reflection coefficients. Consequently I only give the result we need in Eqn. 35 and refer the reader to Ref.~30 for details.

Regarding the comment "the case of zero temperature, I should notice that strictly speaking, one has to describe the behavior of the RHP solution in the points of non-analyticity, otherwise the solution to the RHP is not unique. If this is not the case for the given RHP, it would be desirable to have some explanation." The behaviour of the RHP near the points of non-analyticity for zero temperature are given in Eqns. 25, 27 and 29.

I thank the referee again for their comments and hope they find my reply helpful.

---

## Round 1 · Referee Report · Anonymous (Referee 2) · 2017-10-15

Strengths

1- New determinant representations for dynamical thermal correlators relevant to nonlinear Luttinger liquid physics 2- Asymptotic analysis

Weaknesses

1- Calculations and logic not so easy to follow 2- Result limited to a simple form for the density matrix

Report

In this manuscript the author considers dynamical thermal correlators in a non linear Luttinger liquid model. The main result is an exact Fredholm determinant formula for correlations of vertex operators in a conformally density matrix, generalizing previous results at zero temperature. This is done using the fact that a certain ratio of such correlators satisfies a nonlinear Schroedinger equation. The large time behavior is then obtained by mapping onto a Riemann-Hilbert problem, which can be solved asymptotically at both zero and finite temperature.

The paper is overall interesting, and some of the results are potentially quite nice. Several aspects can (and should) be improved however, especially regarding the writing of the manuscript. As it is several statements are cryptic, and should be supplemented by calculations. The logic behind some of the derivations could be better explained also, it takes the reader quite some effort to determine whether a given sentence is a side comment on the previous equation, a proof, or a general statement that will be useful next.

In section 3, the author should really make an effort to make the derivations easier to reproduce, especially bearing in mind that asymptotic analyses to Riemann-Hilbert problems are sometimes difficult to follow. Some more physical discussions of the final results would be desirable also.

The paper might be publishable, provided these concerns are addressed. Below is a list of more minor comments, some of which are still related to my points above.

1) The introduction is too short and narrow as it is. The author should spend more time introducing the problem, what motivates it, while discussing the literature. 2) Page 2, after equation (3), "The bosonization identity [...]". Add a reference. 3) Page 3. "It is easy to include them to any result [...]". How so? 4) Page 3. What is $m$ in equation (5)? 5) Page 4. "To this end it's" --> "To this end it is" 6) Page 5, around equation (5). Mention that $\sigma_3$ is a Pauli matrix. 7) Page 5, before equation (16). Write $\theta(k)$ instead of $\theta$ to avoid any ambiguity in the following. 8) Page 5, after equation (17). "The proof is to observe [...]". The proof of what? 9) Page 6. I do not understand what is meant exactly at the end of the first paragraph. 10) Page 6, "tends to the identity at infinity". When $z$ goes to infinity would be slightly more precise. 11) Page 7, section 3.2. The discussion at the beginning of the section could be made clearer I think. Also, some of the sentences in this paragraph read awkwardly. 12) Page 9, equation (33). The author should explain more precisely what are the contours i, II, III, IV, V and VI. Same comment in the caption to figure 1. 13) Page 11, equations (39), (40), (41). These results deserve a proper discussion, not only in the conclusion, especially given the fact that everything appears to be dominated by the zero temperature Fermi point even at finite temperature. 14) Appendix A. A few lines of introduction, explaining what is being computed in this appendix, would help. The sentence "As always we consider $x$ having a positive imaginary infinitesimal" reads awkwardly.

Requested changes

Improve the writing and the derivations, in particular in section 3.
Address my other minor comments.

---

## Round 1 · Referee Report · Anonymous (Referee 3) · 2017-11-2

Strengths

See report

Weaknesses

see report

Report

The paper "Long time thermal asymptotics of nonlinear Luttinger liquid from inverse scattering" by Tom Price
provides a Fredholm determinant based representation for the dynamical two point functions of vertex operators at finite temperature.
This representation is then used to characterisation of these correlators in the long time regime at finite and zero temperature.

In itself, the paper is rather interesting as it brings together techniques stemming from quite distant fields so as to
obtain physically interesting results for the correlators of a nonlinear Luttinger Liquid.
Although the techniques used by the author are well known in their respective fields, what makes the originality of the paper
is the fact of putting these thematically distant pieces together. Unfortunately, the paper is written in a very cryptic way what makes the reasoning hard to follow.
Since the paper develops concepts borrowed from various fields, and people are seldom experts in all of these, I do not believe that
saving space by giving less details is an option.

Below I list the various comments and suggestions and I am ready to recommend the paper for publication as soon as these are met.

Abstract

"The asymptotics display exponentially small corrections in temperature"
It should be made clear which regime $T->0$, $T-> \infty$ is considered?

I Introduction

I disagree with the author's way of presenting things: "For integrable theories at zero temperature the mobile impurity phenomenology can give exact results". First of all, the method, without further modifications, only catches leading asymptotics hence does not produce exact results. Furthermore, the exact results stemming from quantum integrable models confirm that the heuristics of the non-linear luttinger liquid are correct, to the leading order.

Independently of that, the work [12] builds on the techniques developed in N.A. Slavnov, "Non-equal time current correlation function in a one-dimensional Bose gas.", Theor. Math. Phys., 82, (1990), 273-282. N.Kitanine, K.K.Kozlowski, J.-M.Maillet, N.A.Slavnov, V.Terras, "On the thermodynamic limit of form factors in the massless XXZ Heisenberg chain.", J. Math. Phys., 50, (2009), 095209
and ibidem, "Thermodynamic limit of particle-hole form factors in the massless XXZ Heisenberg chain.", J. Stat. Mech. : Th. and Exp., 1105, (2011), P05028.

Instead of [20], I would find it more appropriate to cite the founding paper of the method "Differential equations for quantum correlation functions.", A.Its, A.Izergin, V.Korepin and N.Slavnov, Int. J. Mod. Physics ,B4, (1990), 1003-1037. This remark also holds in the core of the paper, above (14) and (18).

I.1

What the author refers to as vertex operator are, in fact, vertex like operators since the zero modes are missing from the definition (and one needs these to ensure conformal properties of transformation of the fields)

below (4): the action of the operators F and $F^{\dagger}$ should be specified.

Also, since one is in finite volume, something should be said here about the quanitifaction of the momenta defining the Fermi surface.

below (5) "It is easy to include them to any results obtained at the end of the calculation."
I would find it useful to have a discussion about how one can include these zero mode and how this alters the results (for instance in a short appendix).

Paragraph below (6): it is not completely clear to me what the author means by "the effect of
dispersion in the thermodynamics should be nonsingular". It would be good to explain the sentence better and
add a reasoning (even a hand waving one) that would support the statement.

Next paragraph: The author should refer here to T.Dorlas, J.Lewis and J.Pulè, "The Yang-Yang thermodynamic formalism and large deviations.", Comm. Math. Phys., 124, Issue 3, (1989), 365-402. where the rigorous, large deviation principle based, approach to the treatment of thermodynamics -in the sense described in that paragraph-, was developed.
Yang-Yang's arguments were only heuristic. There are also other approaches, based on explicit resummations of the form factor series, which allow one to deal with thermodynamics of free fermion systems. I find that it would be relevant to cite them here, for instance V.Korepin and N.Slavnov, "The time dependent correlation function of an impenetrable Bose gas as a Fredholm minor I.", Comm. Math.Phys., 129, 1, (1990), 103-113. (dynamical correlators of impenetrable Bosons at zero temperature, first appearance of the method) or F.Colomo, A.Izergin, V.Korepin and V.Tognetti, "Correlators in the Heisenberg XX0 chain as Fredholm determinants.", Phys. Lett., A 169, (1992), 237-247 (extension to the finite temperature case).

(7) To keep consistence with the rest of the paper, one should use the notation $G_{\eta}(x,0)$ etc.

II Fredholm determinant

This section contains the proof of the determinant representation in the dynamic and finite temperature case.
The proof for the $t=0$ representation is given in Appendix A (see the comments related to that part later on). However, for the time dependent case, the proof resumes to the sentence "The proof is to observe [20] that the traces of the resolvent (15) of integrable kernels with time dependence of the form (12) satisfy NLSEs (8), so it remains only to check the value at t=0".
Although I agree with the statement, one should provide more computational details of the statement.

(11) I believe that writing the kernel in terms of a scalar product, as it is customary in the Riemann-Hilbert problem community would ease the reading.

(13) one could already use the notation $\mathcal{G}^2$ introduced in (16) so as to shorten the formula. This would be more consistent (43).

(15) It is more natural, especially regarding to the effective use of the Riemann--Hilbert analysis, to express (15) in terms of the entries of the solution to the RHP associated with the Fredholm determinant of interest.

I disagree with the statement " It appears consistent with the results of Ref. [5] that the dynamical phase evolves with the energy relative to the ground state at zero temperature, not relative to a “representative state” with occupation
$\theta$(k)." This is a specific feature of free fermion equivalent models which ceases to be true when interactions are turned on.
See, for instance, the dressing effect of the bare momentum in the static case that was shown to appear in
K.Kozlowski, J.-M.Maillet and N.Slavnov, "Long-distance behavior of temperature correlation functions of the quantum one-dimensional Bose gas.", J. Stat. Mech., (2011), P03018. A similar dressing effect arises in the dynamical case as well, with the dressing now concerning also the energy of excitations. I refer to

A.R.Its, N.A.Slavnov, "On the Riemann-Hilbert approach to the asymptotic analysis of the
correlation functions of the Quantum Nonlinear Schr\"{o}dinger equation. Non-free fermion case.", Theor. Math. Phys., 119:2, (1990),541-593.

N.A.Slavnov,"Integral equations for the correlation functions of the quantum one-dimensional Bose gas.",Theor. Math. Phys., 121, (1999), 1358-1376.

"F.Göhmann, M.Karbach, A.Klümper, K.K.Kozlowski, J.Suzuki, "Thermal form-factor approach to dynamical correlation functions of integrable lattice models", Cond.mat 1708.0406

Concerning the last paragraph, I believe that a direct proof is possible, probably by adapting in an appropriate way the form factor summation techniques pioneered in the mentioned
papers of Korepin and Slavnov and Colomo, Izergin, Korepin and Tognetti. Also, the functions appearing in the end of the paragraph are, again, missing their t dependence.

III Large x,t asymptotics

The treatment of RHP is extremely sparse, no precise formulation of the RHP is given (asymptitics, type of boundary values etc). It is very hard to follow the reasoning, even for experts. Definitely much more details should be given and the presentation improved/clarified.

There is a non uniformness in the notation for the identity matrix. cf (19), just below it and in (20), where it is denoted by 1.

The transformation (21) should refer to the original work where this idea was developed (Its, Izergin, Korepin, Varzugin "Large
time and distance asymptotics of field correlation function of impenetrable bosons at finite temperature ", Physica D 54 (1992) 351-395)

(24) One should provide more details on where this formula comes from

Section 3.2

(26)-(29): Ref. [15] studied a local analogue of the RHP studied by the author in the present paper. The solution was given explicitly in terms of confluent hypergeometric functions. I believe that this is more efficient in respect to extracting the large-time asymptotics from the parametrix then the Cauxhy integral representation obtained by the author. Indeed, the latter has non-uniform asymptotics (depending on whether $z$ is close to $k_{*}$ or far away) and establishing these demands an extra amount of work.

The statement "Since $m_>$ is traceless" is strange in that it follows from (27) or (29) that the trace of that matrix is $2$. Finally, the statement "the only contribution to $\partial_x G_{\eta}$ is from the Fermi point" is incorrect. It does probably hold in what concerns the leading asymptotics of $G_{\eta}$, but subleading ones -despite the upper or lower triangular structure-
do contain both contributions from the Fermi boundary and from the saddle-point $k_{*}$. Also, the saddle-point $k_{*}$ should contribute to the leading asymptotics in the case of $L_{\eta}$

Equation (30) needs definitely much more justification that the few words that precede it.

"At $\eta=0$ they reduce to the asymptotics of the free fermion". The model at general $\eta$ can still be understood as a free fermion model. The sole difference being that the quantisation conditions between the ground state and the excited states should be slightly modified. This results in a shift function proportional to $\eta$.

Section 3.3

I found the presentation around the 2nd paragraph rather obscure. More details should be given. Also, in this analysis the role played by the poles of the Fermi weight should be explained (and treated) much better. Finally, the role of the $t$ dependent correction in (3.4) should also be discussed precisely.

"All is needed is its residue at infinity, which we already know from Luttinger liquid theory, Eqn. (7)." I do not believe that such a statement is an option when carrying out a Riemann-Hilbert analysis.

Although [30] contains the RHP solved by the parabolic cylinder functions, I believe that the correct reference here would rather be
A.R. Its, "Asymptotics of solutions of the nonlinear Schrodinger equation and isomonodromic deformations of systems of linear differential equations", Soy. Math. Dokl., 24, No.3, 452-456 (1981).

Conclusion

"An extension to models with interacting spectra such as Lieb–Liniger or XXZ, rather than the free fermion
spectra considered here, is required to understand the connection of the dressed energy in the
thermodynamic Bethe ansatz and the oscillations in the large t asymptotics"
The author should check, , the three references mentioned above where this problem was addressed.

Appendix A

A.1

The regularisation prescription for $x$ should be better discussed.
Also, I would find more reasonable to turn the proof upside down, ie start from (49) and then going "up" arrive to (42).

A.2

The proof should contain much more details and some explanation of the rules that are used so as to get the result.

Requested changes

See report

---

## Editorial Decision

awaiting_resubmission